# Treatment-Related Late Adverse Events in Childhood Cancer Survivors of Mexico: A Cross-Sectional Study

**José Fernando Pérez-Franco [1], Gabriela Hernández-Pliego [2], Jocelyn Jacobo-Mendoza [3], Vanessa Karina Martínez-Lara [4], Luis Enrique Juárez-Villegas [2], Patricia Clark [1] and Jessica Liliana Vargas-Neri [3,*]**

[1] Clinical Epidemiology Research Unit of Hospital Infantil de México Federico Gómez | Faculty of Medicine of Universidad Nacional Autónoma de México, Mexico City 06720, Mexico

[2] Department of Hemato-Oncology of Hospital Infantil de México Federico Gómez, Mexico City 06720, Mexico

[3] Clinical Epidemiology Research Unit of Hospital Infantil de México Federico Gómez | Faculty of Chemistry of Universidad Nacional Autónoma de México, Mexico City 06720, Mexico

[4] Clinical Epidemiology Research Unit of Hospital Infantil de México Federico Gómez, Mexico City 06720, Mexico

* Correspondence: lilianavargasneri@gmail.com

**Abstract:** Late adverse events (LAEs) are an important cause of illness and disability in childhood cancer survivors (CCSs) and increase the risk of mortality. The aim of this cross-sectional study was to describe the frequency and severity of treatment-related LAEs in Mexican CCSs. The study period was between September 2018 and April 2019. We tested a sample of 82 CCSs at the Hospital Infantil de México Federico Gómez. We considered an LAE to be any medical effect related to treatment after ending cancer therapy. All LAEs were classified according to severity (using the grades of Common Terminology Criteria for Adverse Events v.5.0), diagnosis and time of occurrence after treatment. The treatment-related LAE frequency was 11.0% (95% CI; 4.2–17.8%). A total of 11 LAEs were identified in nine patients. Slightly over half of the patients were male (54.9%). The most frequent diagnosis was acute lymphoblastic leukemia (45.1%). The body systems involved in LAEs were the endocrine (55.6%), neurological (22.2%), auditory (11.1%) and renal (11.1%) systems. Obesity was the most frequent LAE (45.4%). Most LAEs were classified as grade 1 and 2 (60%). The median follow-up was 6.5 years. The odds ratio was used as a measure of association to identify characteristics associated with the LAEs. We identified that the age at diagnosis (OR = 0.71, 95% CI, 0.51–0.99; $p = 0.046$) and chemotherapy-only group (OR = 0.03, 95% CI, 0.00–0.86, $p = 0.040$) were associated with LAEs. This is the first study that describes the frequency and severity of LAEs in Mexican childhood cancer survivors.

**Keywords:** childhood cancer; cancer survivors; late adverse events; chemotherapy; surveillance

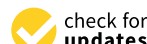



## 1. Introduction

There are approximately 300,000 childhood cancer survivors (CCSs) in the United States (U.S.) [1]. Approximately 80% of these patients survive more than five years after treatment [2]. In the case of low- and middle-income countries (LMICs), such as Mexico, the childhood cancer survival rate is approximately 57% [3,4]. The main causes for this are late diagnoses, limited access to treatment, a lack of treatment adherence and disease relapse [5].

Continued advances in pediatric cancer therapy are the reason for the number of increased childhood cancer survivors, and this increases disease-free life expectancy and treatment-related toxicity [6,7].

Chemotherapy has a narrow therapeutic index, resulting in an increase in adverse events (AEs), which may appear during treatment or years after the end of treatment [8]. These AEs can seriously impair survivors' overall health or be potentially life-threatening [7,9].

The most common body systems affected by treatment-related late AEs (LAE) reported in the literature are the endocrine, cardiovascular, pulmonary, neurocognitive, musculoskeletal, renal and auditory systems [10,11]. Approximately 57% of anthracycline-treated childhood cancer survivors have asymptomatic signs of cardiotoxicity that often progress over time, which is the main cause of morbidity and mortality after cancer recurrence and secondary malignancies [12,13].

There may be prolonged latency between exposure to treatment and AE manifestation, such that the first-contact physician does not suspect or report it [14–16]. It has been documented that, up to 30 years after finishing treatment, childhood cancer survivors experience LAEs that can become disabling [17]. LAEs of therapy are defined as acute or chronic diseases appearing after treatment ends. These LAEs can be classified using the Common Terminology Criteria for Adverse Events (CTCAE), a descriptive terminology that allows us to classify AEs by severity. The CTCAE uses grades from 1 to 5, with unique clinical descriptions of the severity of each AE [18]. There are particular factors that increase the risk of developing LAEs, such as age at diagnosis, sex, cumulative chemotherapy dose, comorbidities and race [19].

Some AEs can be identified early and treated without leaving sequelae, and others may persist in adulthood as chronic diseases or contribute to the progression of other diseases, such as LAEs [11]. With respect to Mexico, there are no reports of epidemiological surveillance of LAEs in CCSs that help evaluate the impact on their quality of life [20]. Therefore, the aim of this cross-sectional study was to describe the frequency and severity of treatment-related LAEs in Mexican CCSs.

## 2. Results

A total of 82 patients were included in this study; Table 1 shows the demographic characteristics. The median age was 6.0 years (IQR 3.0–13.0 years). Slightly over half of the patients were male (54.9%). The most frequent diagnosis was acute lymphoblastic leukemia (45.1%), followed by lymphoma (14.6%) and germ cell tumors (14.6%). Chemotherapy (CT) alone was used in 37 (45.1%) patients, and 16 (19.5%) had undergone CT with radiotherapy (19.5%). There were 17 (20.7%) patients treated with CT and surgery and 12 (14.6%) patients treated with CT, surgery and radiotherapy. The median follow-up was 6.5 years (IQR 3.0–12.0 years). Patient outcome was considered in two categories: survival or death. Only 4 (4.9%) patients died, but the cause was unknown. There was a difference between the presence or absence of LAE in patients who received chemotherapy only ($p = 0.030$).

Table 2 shows the most common body systems affecting treatment-associated LAEs in CCSs. Of the 82 survivors, 9 (11.0%) had one or more LAEs, and 73 (89.0%) had no LAEs. A total of 11 LAEs were identified. The treatment-related LAE frequency was 11.0% (95% CI, 4.2–17.8%). When we evaluated the body system, five patients presented LAEs of the endocrine system (55.6%), two patients presented LAEs in the neurological system (22.2%), and one patient presented LAEs in each of the auditory (11.1%) and renal (11.1%) systems. Obesity was the most frequent LAE (45.4%). Anthropometric data (weight and height) were obtained prior to treatment in these patients. The mean of the initial (prior treatment) body mass index (BMI) of these patients was 21.7 kg/m$^2$. This value helped us to have a temporary association of obesity with the treatment. Only two patients had LAEs affecting two body systems at the same time. One of the CCSs had the endocrine and neurological systems affected, and the other had the endocrine and auditory systems affected. No childhood cancer survivor had more than two body systems affected.

**Table 1.** Demographic characteristics of the included patients.

| Characteristics | Study Population *n* = 82 | Patients with LAEs *n* = 9 | Patients without LAEs *n* = 73 | *p*-Value |
|---|---|---|---|---|
| Age at diagnosis (years) [median and IQR] | 6.0 (3.0–13.0) | 5.0 (2.5–12.5) | 6.0 (3.0–13.0) | 0.666 |
| Sex | | | | |
| Male (*n*, %) | 45 (54.9) | 3 (33.3) | 42 (57.5) | 0.287 |
| Weight (kg) [median and IQR] | 25.0 (13.9–49.6) | 27.0 (14.5–66.0) | 24.0 (13.8–47.3) | 0.369 |
| Height (cm) [median and IQR] | 113.3 (88.0–152.3) | 100.0 (75.5–143.6) | 114.0 (89.0–152.5) | 0.509 |
| Diagnosis | | | | |
| Leukemia (*n*, %) | 37 (45.1) | 3 (33.3) | 34 (46.6) | 0.349 |
| Lymphoma (*n*, %) | 12 (14.6) | 2 (22.2) | 10 (13.7) | 0.392 |
| Germ cell tumors (*n*, %) | 12 (14.6) | 1 (11.1) | 11 (15.1) | 0.608 |
| Central nervous system tumor (*n*, %) | 5 (6.1) | 1 (11.1) | 4 (5.5) | 0.450 |
| Rhabdomyosarcoma (*n*, %) | 3 (3.7) | 0 (0.0) | 3 (4.1) | 1.00 |
| Retinoblastoma (*n*, %) | 3 (3.7) | 0 (0.0) | 3 (4.1) | 1.00 |
| Wilms tumor (*n*, %) | 3 (3.7) | 1 (11.1) | 2 (2.7) | 0.298 |
| Neuroblastoma (*n*, %) | 2 (2.4) | 1 (11.1) | 1 (1.4) | 0.209 |
| Ewing's sarcoma (*n*, %) | 2 (2.4) | 0 (0.0) | 2 (2.7) | 1.00 |
| Osteosarcoma (*n*, %) | 1 (1.2) | 0 (0.0) | 1 (1.4) | 1.00 |
| Hepatoblastoma (*n*, %) | 1 (1.2) | 0 (0.0) | 1 (1.4) | 1.00 |
| Langerhans cell histiocytosis (*n*, %) | 1 (1.2) | 0 (0.0) | 1 (1.4) | 1.00 |
| Treatment | | | | |
| Chemotherapy only (*n*, %) | 37 (45.1) | 1 (11.1) | 36 (49.3) | **0.030 *** |
| Chemotherapy + radiation (*n*, %) | 16 (19.5) | 4 (44.4) | 12 (16.4) | 0.068 |
| Chemotherapy + surgery (*n*, %) | 17 (20.7) | 2 (22.2) | 15 (20.5) | 0.598 |
| Chemotherapy + surgery + radiation (*n*, %) | 12 (14.6) | 2 (22.2) | 10 (13.7) | 0.392 |
| Follow-up (years) [median and IQR] | 6.5 (3.0–12.0) | 7.0 (3.0–13.0) | 6.0 (3.0–12.0) | 0.778 |
| Patient outcome | | | | |
| Survivors (*n*, %) | 78 (95.1) | 8 (88.9) | 70 (95.9) | 0.378 |
| Death (*n*, %) | 4 (4.9) | 1 (11.1) | 3 (4.1) | |

To compare the groups of patients with LAEs and patients without LAEs, sex, diagnosis, treatment and patient outcome were evaluated with Fisher's exact test. To compare age at diagnosis, weight, height and follow-up, the Mann–Whitney U test was used. Bolded font and (*) indicate statistically significant values with *p* < 0.05. LAEs: Late adverse events. IQR: Interquartile range.

**Table 2.** Frequency of late adverse events (LAEs) by body system.

| Body System | LAEs (*n*, %) | | Patients (*n*, %) |
|---|---|---|---|
| Endocrine | 6 (54.5) | | 5 (55.6) |
| Obesity | | 5 (45.4) | |
| BMI 25–29.9 kg/m$^2$ | | 3 (27.2) | |
| BMI 30–39.9 kg/m$^2$ | | 2 (18.2) | |
| Thyroid function, low (hypothyroidism) | | 1 (9.1) | |
| Neurological | 2 (18.2) | | 2 (22.2) |
| Cognitive disturbance | | 1 (9.1) | |
| Seizure | | 1 (9.1) | |
| Auditory | 2 (18.2) | | 1 (11.1) |
| Hearing loss | | 1 (9.1) | |
| Otitis, middle ear (non-infectious) | | 1 (9.1) | |
| Renal | 1 (9.1) | | 1 (11.1) |
| Urinary electrolyte wasting (renal tubular acidosis) | | 1 (9.1) | |
| Total | | 11 (100.0) | 9 (100.0) |

BMI: Body mass index. LAE = Late adverse event.

We classified LAEs according to the CTCAE v5.0. Grade 2 or moderate LAEs were the most frequent (54.5%), limiting age-appropriate instrumental activities of daily living. CCSs with a leukemia diagnosis had one (9.1%) LAE classified as grade 4, which equates to life-threatening consequences. None of the patients had grade 5 or death-related LAEs (Table 3).

**Table 3.** Frequency of severity grade of LAEs according to CTCAE by diagnosis.

| Diagnosis | Severity Grade | | | | |
|---|---|---|---|---|---|
| | 1 | 2 | 3 | 4 | 5 |
| Leukemia (*n*, %) | 1 (50.0) | 1 (16.7) | 1 (50.0) | 1 (100.0) | 0 (0.0) |
| Lymphoma (*n*, %) | 0 (0.0) | 2 (33.3) | 0 (0.0) | 0 (0.0) | 0 (0.0) |
| Other types of cancer (*n*, %) | | | | 0 (0.0) | 0 (0.0) |
| Central nervous system tumor (*n*, %) | 1 (50.0) | | | | |
| Germ cell tumors (*n*, %) | | 1 (16.7) | | | |
| Rhabdomyosarcoma (*n*, %) | | 1 (16.7) | | | |
| Wilms tumor (*n*, %) | | 1 (16.7) | | | |
| Neuroblastoma (*n*, %) | | | 1 (50.0) | | |
| Total *n* = 11 | 2 (18.2) | 6 (54.5) | 2 (18.2) | 1 (9.1) | 0 (0.0) |

LAE = Late adverse events. CTCAE = Common Terminology Criteria for Adverse Events.

LAEs can appear after cancer therapy ends. Figure 1 shows that CCSs with a follow-up of 7 to 9 years presented a higher number of LAEs affecting three different body systems: the endocrine, neurological and renal systems. However, these may appear from 1 year after the end of cancer therapy until 18 years later. Late endocrine system adverse events were present in all time periods.

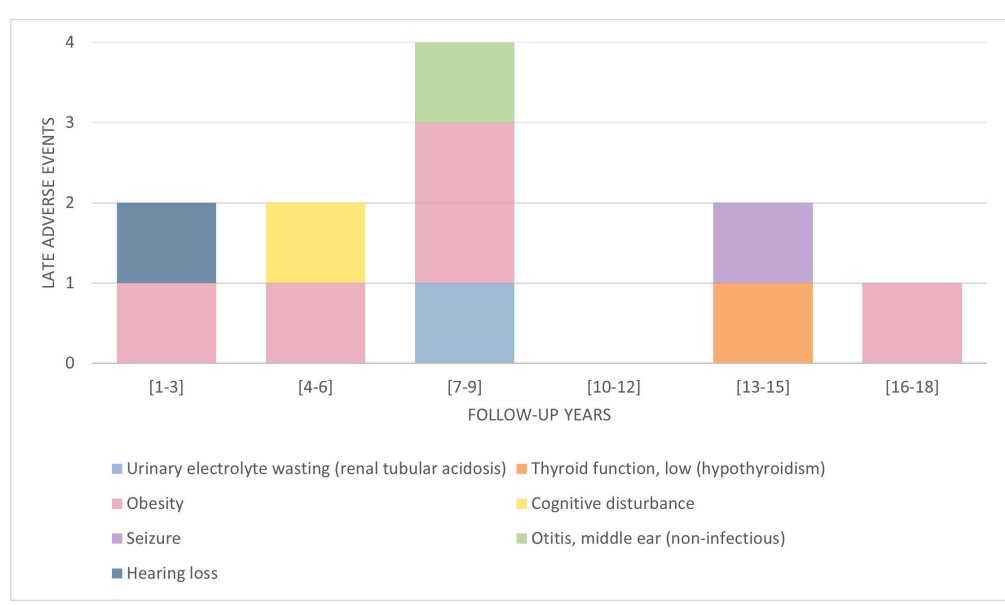

**Figure 1.** Late adverse events by follow-up years.

To identify characteristics associated with LAEs, we used the odds ratio (OR) as a measure of association. In the bivariate analysis, the treatments of chemotherapy only (OR = 0.13, 95% CI, 0–01–1.08; *p* = 0.030) and chemotherapy + radiation (OR = 4.07, 95% CI, 0.95–17.40; *p* = 0.045) were statistically significant (Table 4).

**Table 4.** Characteristics associated with late adverse events.

| Variable | Patients with LAEs $n = 9$ | Patients without LAEs $n = 73$ | Unadjusted OR (95% CI) | *p*-Value | Adjusted OR (95% CI) | *p*-Value |
|---|---|---|---|---|---|---|
| Age at diagnosis | ----- | ----- | ----- | ----- | 0.71 (0.51–0.99) | **0.046 *** |
| Weight | ----- | ----- | ----- | ----- | 1.09 (0.98–1.21) | 0.096 |
| Height | ----- | ----- | ----- | ----- | 1.00 (0.93–1.06) | 0.962 |
| Sex | | | | | | |
| Female | 6 | 31 | 2.71 (0.63–11.69) | 0.169 | 4.54 (0.83–24.81) | 0.081 |
| Diagnosis | | | | | ----- | ----- |
| Leukemia | 3 | 34 | 0.57 (0.13–2.47) | 0.451 | | |
| Lymphoma | 2 | 10 | 1.80 (0.33–9.92) | 0.495 | | |
| Germ cell tumor | 1 | 11 | 0.705 (0.08–6.20) | 0.751 | | |
| Central nervous system tumor | 1 | 4 | 2.16 (0.21–21.73) | 0.505 | | |
| Rhabdomyosarcoma | 0 | 3 | 0.89 (0.82–0.96) | 0.536 | | |
| Retinoblastoma | 0 | 3 | 0.89 (0.82–0.96) | 0.536 | | |
| Wilms Tumor | 1 | 2 | 4.44 (0.36–54.56) | 0.207 | | |
| Neuroblastoma | 1 | 1 | 9.00 (0.51–158.17) | 0.074 | | |
| Ewing's sarcoma | 0 | 2 | 0.89 (0.82–0.96) | 0.615 | | |
| Osteosarcoma | 0 | 1 | 0.89 (0.82–0.96) | 0.724 | | |
| Hepatoblastoma | 0 | 1 | 0.89 (0.82–0.96) | 0.724 | | |
| Langerhans cell histiocytosis | 0 | 1 | 0.89 (0.82–0.96) | 0.724 | | |
| Treatment | | | | | | |
| Chemotherapy only | 1 | 36 | 0.13 (0.01–1.08) | **0.030 *** | 0.03 (0.00–0.86) | **0.040 *** |
| Chemotherapy + radiation | 4 | 12 | 4.07 (0.95–17.40) | **0.045 *** | 1.73 (1.20–15.30) | 0.621 |
| Chemotherapy + surgery | 2 | 15 | 1.10 (0.21–5.87) | 0.907 | 0.26 (0.02–3.24) | 0.293 |
| Chemotherapy + surgery + radiation | 2 | 10 | 1.80 (0.33–9.92) | 0.495 | Reference | |
| Patient outcome | | | | | | |
| Survivors | 8 | 70 | ----- | ----- | 0.15 (0.01–2.93) | 0.211 |

CI = confidence interval, LAE = late adverse events, OR = Odds Ratio. Odds ratio adjusted for age at diagnosis, sex, weight, height, treatment, and patient outcome. Bolt font and (*) indicates statistically significant values with $p < 0.05$. LAEs: Late adverse events.

In a model of logistic regression (with OR adjusted for age at diagnosis, sex, weight, height, treatment and patient outcome), the characteristics of age at diagnosis (OR = 0.71, 95%CI, 0.51–0.99; $p = 0.046$) and the treatment of chemotherapy only (OR = 0.03, 95%CI, 0.00–0.86; $p = 0.040$) were associated with LAEs. (Table 4).

We calculated post hoc statistical power with our sample size. This study had a power of 79.7%. In our study, the point estimate was 11% of LAEs, which can range from 4.2 to 17.5% with a precision of 7.4%.

## 3. Discussion

The LAEs found in our study involved the endocrine, neurological, auditory, and renal systems. This cross-sectional study allowed us to determine the characteristics of LAEs presented in this population.

The fact of having survived cancer already indicates success because the patients have already been cured. However, we must not forget that these patients may present sequalae associated with cancer therapy months or years after finishing treatment. Children are more susceptible to developing AEs than the adult population because they have an immature pharmacokinetic and pharmacodynamic profile that generates differences in the absorption, distribution, metabolism and elimination of drugs [21–23]. It has been

estimated that approximately 75% of CCSs have suffered at least one LAE after finishing treatment, and almost 40% of them have had one severe or life-threatening LAE [24].

We found a treatment-related LAE frequency of 11.0% (95% CI; 4.2–17.8%). Most LAEs were classified as grade 1 and 2 (60%), the same as that found by Geenen M. et al., but the difference with our study was that we did not find deaths due to an LAE [24]. We reported CCS deaths, and the causes of these deaths were unknown. Therefore, we could not associate these deaths with the treatment. However, other studies have reported deaths due to LAEs [25,26]. Differences in cancer diagnosis and treatment modalities used in other countries could explain the differences in CCS outcomes [7].

LAEs were identified across all treatment regimens. CCSs with life-threatening or disabling LAEs were more frequent in the chemotherapy–radiotherapy group. The endocrine system (56%) was the most frequently affected body system. Similar to Wilson C. et al., we found that 45% of CCSs were obese according to CTCAE [27]. This instrument is limited in that it does not distinguish the nutritional status of children and is generalized to adults.

Other studies have reported that obesity risk in CCSs is influenced by the long-term use of high-dose glucocorticoids during treatment and additional factors such as radiotherapy [27]. The use of this kind of drug represents a risk factor associated with the proliferation and maturation of visceral adipocytes [28]. In the case of Mexican CCSs, they present other risk factors in adulthood, such as a sedentary lifestyle (29%), alcoholism (21.7%), dyslipidemia (19.5%), hypertension (18.4%) and smoking (11.4%), which increase the risk of developing obesity and metabolic syndrome [29]. We consider that the frequency of obesity found in our study was lower than the prevalence of obesity in Mexico because the CCSs, despite being exposed to obesogenic factors during surveillance, were patients who, during treatment, presented a loss of appetite, changes in lifestyle and fear of having other diseases. We understand that obesity as an LAE has many risk factors that increase as surveillance time increases and, therefore, cannot be associated only with treatment. It is an area of opportunity for our hospital to implement other diagnostic tests to be performed periodically to evaluate whether the treatment has more association with obesity.

In another study, Livinalli A. et al. found that approximately 40% of CCSs have LAEs in the cardiovascular system, the most prevalent of which is mitral valve alteration [11]. We did not find any, and we hypothesized that these AEs do not occur frequently in our population. In Mexico, in most protocols that include anthracycline treatment, the cumulative dose is lower than 300 mg/m$^2$ [30]. In addition, dexrazoxane, an iron chelator, reduces early myocardial injury during anthracycline treatment [30,31]. However, approximately 57% of CCSs may present asymptomatic signs of cardiotoxicity up to 6.2 years after anthracycline treatment completion [13]. These cardiotoxicity signs can only be detected via echocardiography. In many hospitals, this procedure is usually not available; therefore, these LAEs are not reported until clinical manifestations appear. Evidence in Mexican patients has shown that the occurrence of anthracycline-induced cardiotoxicity depends on clinical and genetic factors. Genetic factors are particularly important, especially when there are protective variants that influence pharmacokinetic processes [30].

Another important point is that we observed that the severity of LAEs was higher when diagnoses were made at a younger age, which reduced the quality of survival through its impact on health and increased disability-adjusted life years (DALYs), meaning the years of life lost due to premature death and years lived with a disability [32].

The median follow-up time in CCSs reported in other studies is 21.1 years, and the median attained age at the end of follow-up is 38.0 years [25]. Unlike them, we found a median follow-up of 7.0 years, and we did not include the adult population. In Mexican children's hospitals, childhood patients receive medical care only until they reach the majority age and then must continue their treatment in another hospital.

Many of these CCSs have LAEs that affect multiple body systems, such as the neurological and auditory systems, which leave permanent sequelae that hinder their social, family, academic and labor functions. In fact, some countries have implemented guidelines and recommendations for the surveillance of LAEs to improve the quality of life of survivors [7].

Moreover, a multidisciplinary center focused on preventing and treating treatment-related LAEs is needed. This is especially important as, to date, long-term follow-up for CCSs is not always well organized. Few pediatric oncology institutions offer follow-up to adult cancer survivors. In Mexico, there is no appropriate transition to the adult system that provides continuous and lifelong follow-up. Surveillance of these LAEs can potentially reduce morbi-mortality if health-preserving early interventions are implemented by specialist physicians [7,33].

Because the aim of this study was to describe the frequency and severity of LAEs in CCSs, we consider that it was a satisfactory exploratory study. In addition, the results of the characteristics associated with LAEs show us that, for each year increase in age, the odds of an LAE are reduced by approximately 30%, and subjects who had only chemotherapy had their odds of LAEs reduced by almost 95% compared to those who received chemotherapy + surgery + radiation. In the logistic regression, the mortality variable is likely playing the role of a confounder because it modifies the estimate. It was included as a confounding variable, and it did not come out as significant. Regardless of whether the patients died, the found effect was maintained.

Nevertheless, we had some limitations in this study, such as the quality of the information, because it depended on clinical records. Therefore, we did not achieve more patients with all the data required. We found that the point estimate was 11% of LAEs, which can range from 4.2 to 17.5% with a precision of 7.4%. Livinalli A et al. reported a prevalence of 27.4% of LAEs and did not offer a 95% CI, to provide an idea of how accurate this was estimated [11]. However, we performed the calculation, and in their sample of 62 patients, their estimate varied from 16.3 to 38.5%. This indicates that both samples were small, but our interest was to describe the frequency by estimating the population proportion of LAEs, whose value can range from 4.23 to 17.7%. Our upper confidence limit (17.7%) coincides with the lower confidence limit of the study of Livinalli A et al. (16.3%) [11]. Another limitation is the time of follow-up due to the Healthcare System transitions. Despite this, we were able to observe follow-ups of more than 12 years. However, these limitations do not decrease the importance of our findings and the need for surveillance of LAEs found in Mexican CCSs. Future studies on CCSs will be needed to further validate these findings and to evaluate the clinical impact of LAEs.

## 4. Materials and Methods

### 4.1. Study Design and Setting and Context

This was a cross-sectional study with active pharmacovigilance of a cohort of childhood cancer survivors at the Hospital Infantil de México Federico Gómez (HIMFG) to describe the frequency and severity of LAEs in this population. HIMFG is a tertiary-care pediatric hospital located in Mexico City with 229 census beds and 120 non-census beds. In 2019, the hospital provided care to more than 11,000 patients from the State of Mexico, Mexico City, Guerrero, Veracruz and Hidalgo states. These patients belong to low-income families, as uncovered by social security systems. According to the Mexican Health System, HIMFG cannot provide care to adult patients.

This study included individuals from the cohort of CCSs at HIMFG. This cohort started in 2018, with 331 eligible individuals that finished the treatment. The individuals received a cancer diagnosis and treatment between October 1995 and August 2018. The cohort of CCSs attempts to address the needs of patients after the completion of treatment.

### 4.2. Participants

The period of this study was between September 2018 and April 2019 in the Hemato-Oncology Department at HIMFG. The sample consisted of 82 patients who met the following inclusion criteria: patients of both genders, patients who had a diagnosis of any type of childhood cancer, patients who had completed oncology therapy and patients who had their treatment administered at HIMFG. The exclusion criteria were patients with ongoing treatment, patients with treatment records in another hospital, patients with incomplete

medical records, patients who did not receive treatment and patients who did not agree to sign the informed consent form. The selection of individuals that participated in this study is shown in Figure 2.

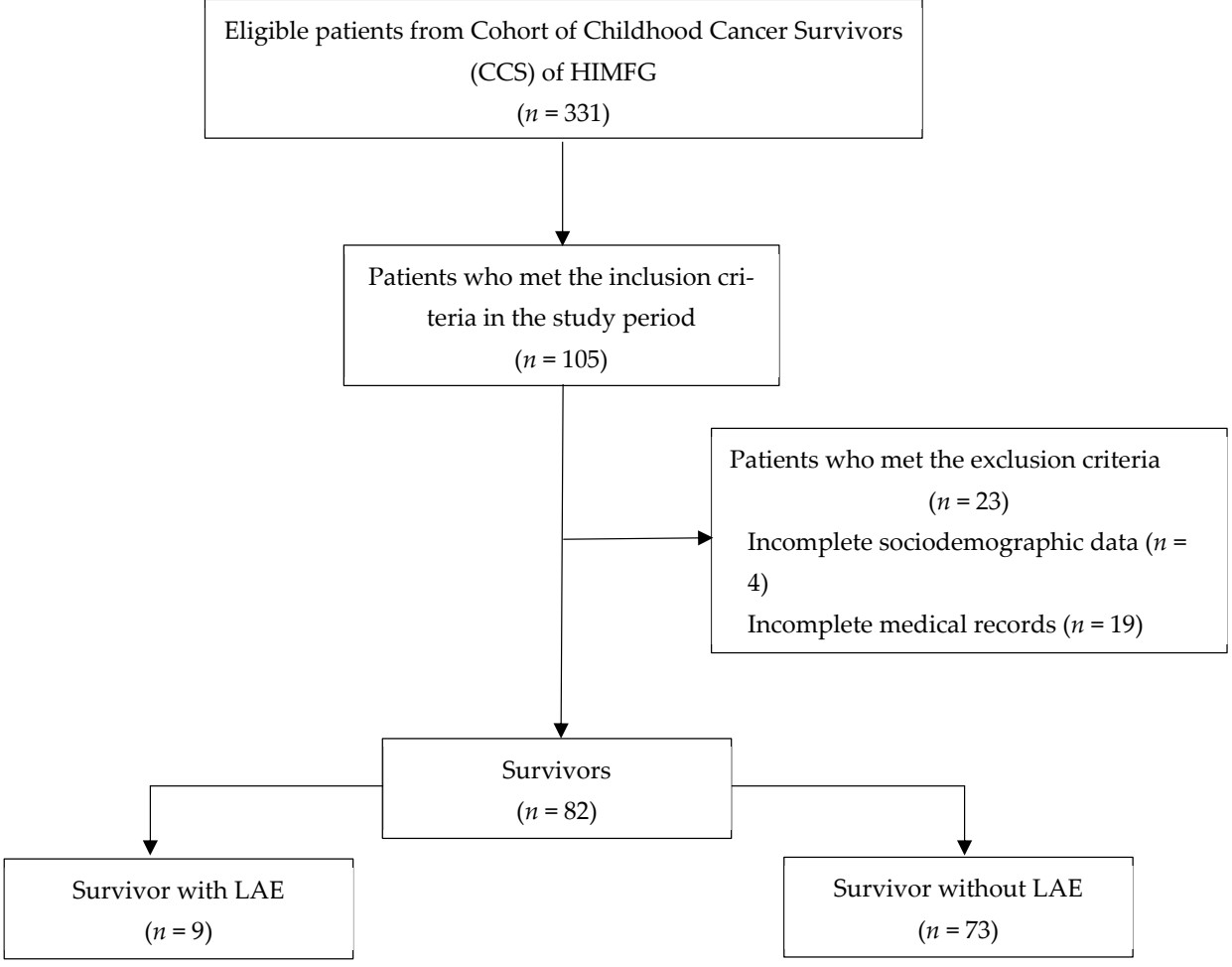

**Figure 2.** Sample composition flowchart. LAE = Late adverse event. CCS = Childhood cancer survivor. HIMFG = Hospital Infantil de México Federico Gómez.

*4.3. Variables and Data Sources*

All information about demographic and clinical characteristics, treatment data and information about LAEs were obtained from medical records. Patients of the CCSs cohort attended surveillance consultations with oncologists. If an LAE was detected, a referral was made to the specialist to confirm and document the adverse event in the medical record. For example, endocrine disorders were confirmed by an endocrinologist with the request of a specific test and evaluations. This information was recorded in medical records and collected for this study. We considered an LAE as any medical effect related to treatment after cancer therapy was finished. Pre-treatment information was obtained from the identified LAEs to confirm that the adverse event was not present prior to exposure to treatment.

The following information was obtained from the medical records: gender; age at diagnosis; diagnosis; weight; size; start and end date of treatment; and treatment, including the regime of chemotherapy, surgery type, radiation data and follow-up at the time of study. The information (sign, symptom, abnormal laboratory or disease) to confirm the LAE was also obtained.

All LAEs were classified according to the Common Terminology Criteria for Adverse Events (CTCAE) v.5.0 [18]. CTCAE is a descriptive terminology that can be utilized for

reported adverse events. A grading (severity) scale is provided for each adverse event term (grade 1 = mild; grade 2 = moderate; grade 3 = severe; grade 4 = life threatening; grade 5 = death related to AE). These LAEs were also classified according to the cancer diagnosis and the time they appeared after the treatment.

*4.4. Statistical Analysis*

Numerical variables with non-normal distributions, such as age, weight, size and follow-up, are presented as medians and interquartile ranges (IQRs). Categorical variables such as sex, diagnosis, treatment and patient outcome are presented as numbers (n) and frequencies (%). The variables were classified according to the presence of LAEs. Clinical and demographic variables were compared between LAE and non-LAE patients using Fisher's exact test and the Mann–Whitney U test. The frequency of LAEs was determined according to the ratio of the number of patients who presented LAEs among the 82 patients included, and it is presented as a number and percentage, with 95% confidence intervals (CI). To identify characteristics associated with LAEs, the odds ratio (OR) was used as a measure of association in a bivariate analysis. The OR of each variable was adjusted for sex, age and treatment using logistic regression. We considered mortality as a confounder. Results were considered statistically significant if *p* values were < 0.05. Analyses were performed using SPSS version 25.0 (IBM Corp, Armonk, NY, USA). We calculated post hoc statistical power with our findings (proportion: 9/82), based on the previous study of Livinalli et al. [11] (proportion: 17/62) using G*Power 3.1.9.7 [34]. The STROBE checklist for cross-sectional study was added to report the study performed with quality (Table S1).

**5. Conclusions**

This is the first study that describes the frequency of LAEs in Mexican childhood cancer survivors. The treatment-related LAE frequency was 11.0% (95% CI; 4.2–17.8%). The body systems involved were the endocrine, neurological, auditory and renal systems. Most of the LAEs were not severe. No cancer survivors died due to LAEs in this study.

**Supplementary Materials:** The following supporting information can be downloaded at: https://www.mdpi.com/article/10.3390/pharma2020015/s1, Table S1: STROBE Statement—Checklist of items that should be included in reports of cross-sectional studies.

**Author Contributions:** Conceptualization, G.H.-P. and J.L.V.-N.; Data curation, G.H.-P. and V.K.M.-L.; Formal analysis, J.F.P.-F., G.H.-P., J.J.-M., V.K.M.-L. and J.L.V.-N.; Investigation, J.F.P.-F., G.H.-P., J.J.-M., V.K.M.-L. and J.L.V.-N.; Methodology, J.F.P.-F., V.K.M.-L. and J.L.V.-N.; Writing—original draft preparation, J.F.P.-F., G.H.-P., J.J.-M., V.K.M.-L. and J.L.V.-N.; Writing—review and editing, J.F.P.-F., G.H.-P., L.E.J.-V., P.C. and J.L.V.-N. All authors have read and agreed to the published version of the manuscript.

**Funding:** This research received no external funding.

**Institutional Review Board Statement:** This study was conducted according to the principles of the Declaration of Helsinki. The protocol was approved by the Research, Ethics and Biosafety Committees at the Hospital Infantil de México Federico Gómez. (Registration code: HIM/2017/126 SSA-1461).

**Informed Consent Statement:** All patients were invited to the study, and written informed consent or assent was obtained from all patients or their parents or legal guardians.

**Data Availability Statement:** The data presented in this study are available on request from the corresponding author (lilianavargasneri@gmail.com).

**Acknowledgments:** We are especially grateful to the specialist physicians of the Hemato-Oncology Department of the Hospital Infantil de México Federico Gómez who attended CCSs. The authors also acknowledge patients and their families for their participation. This would not have been possible without all the support given.

**Conflicts of Interest:** The authors declare no conflict of interest.

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
