# Peer review of "Treatment-Related Late Adverse Events in Childhood Cancer Survivors of Mexico: A Cross-Sectional Study"

_2813-0618, doi:10.3390/pharma2020015_

Round 1
Reviewer 1 Report
Comments and Suggestions for Authors
The authors report on the incidence of late adverse events among childhood cancer survivors (CCS) at a single academic children's hospital in Mexico City. This is an important topic and challenging to address -- it can be extremely difficult to adjudicate whether a medical issue is truly an adverse event related to childhood cancer therapy, or if this is an unrelated de novo effect.
My primary concern is that the authors do not definitively demonstrate that the issues they are identifying are truly related to childhood cancer therapy. In particular, they report 5 patients with an LAE of obesity-- how do they know that obesity is truly an LAE of therapy? With childhood obesity rates around 30% in Mexico, could this not simply be spontaneous obesity? Is there obesity among the CCS cohort that is not considered an LAE? If only 5/82 in the CCS cohort are obese, that would actually be lower than expected based on the spontaneous incidence of obesity. Given that obesity was by far the most common "LAE" reported, I'm concerned about the robustness of the paper as a whole.
Secondly, in table 1, the authors report on demographic characteristics of patients with and without LAE and the study population as a whole. I do not believe the p-values here are corrected for the fact that 16+ hypotheses were tested. They should incorporate some correction (e.g. Benjamini-Hochberg) for false discovery rate in the setting of 16+ hypotheses being tested. My suspicion is that the corrected p-value for chemotherapy only will be above 0.05.
Author Response
Point 1: The authors report on the incidence of late adverse events among childhood cancer survivors (CCS) at a single academic children's hospital in Mexico City. This is an important topic and challenging to address -- it can be extremely difficult to adjudicate whether a medical issue is truly an adverse event related to childhood cancer therapy, or if this is an unrelated de novo effect.
Response 1: We appreciate your comment. We agree with you about the difficulty in adjudicating an adverse event with chemotherapy, so we explored the issue through this cross-sectional study. The study aimed to describe the frequency and severity of late adverse events in cancer survivors. To explore the association of events with therapy, we improved the explanation of the methodology. We mentioned that the LAEs detected did not occur before the administration of treatments (page 3, lines 106-107). As noted, it is not easy to adjudicate causality, so what we wanted to do through this manuscript was to explore the events presented in CCSs, not at the level of causality but at the level of association. Through this study, we reported the frequency of LAEs with its 95% confidence interval (page 3, lines 125-127).
To clarify the objective of the study, we added the type of study in the title (page 1, line 3). We described the setting, context, participants, variables, and data sources (page 2-3). In addition, we added a table describing the characteristics associated with LAEs (page 8).
Point 2: My primary concern is that the authors do not definitively demonstrate that the issues they are identifying are truly related to childhood cancer therapy. In particular, they report 5 patients with an LAE of obesity-- how do they know that obesity is truly an LAE of therapy? With childhood obesity rates around 30% in Mexico, could this not simply be spontaneous obesity? Is there obesity among the CCS cohort that is not considered an LAE? If only 5/82 in the CCS cohort are obese, that would actually be lower than expected based on the spontaneous incidence of obesity. Given that obesity was by far the most common "LAE" reported, I'm concerned about the robustness of the paper as a whole.
Response 2: Response 2: Thank you very much for your comment. We agree with you that we cannot do a causality analysis and much less for obesity which has many risk factors in our population. Through this study, we looked for a temporal association. That is, we corroborated that the outcome of LAE did not occur before treatment. For obesity, we added the BMI data patients with LAE had before treatment (page 6, lines 165-168). In addition, there was no obesity among CCSs that was not considered a LAE.
We understand that obesity as LAE has many risk factors that increase as surveillance time increases and, therefore, cannot be associated only with treatment. It is an area of opportunity for our hospital to implement other diagnostic tests to be performed periodically to evaluate whether the treatment has more association with obesity. Finally, we consider that the frequency of obesity found in our study was lower than the prevalence of obesity in Mexico because, despite being exposed to a lot of obesogenic factors during surveillance, patients during treatment presented loss of appetite, changes in lifestyle and fear of having other diseases This was added in the discussion part of the manuscript (page 9, lines 241-245).
Point 3. Secondly, in table 1, the authors report on demographic characteristics of patients with and without LAE and the study population as a whole. I do not believe the p-values here are corrected for the fact that 16+ hypotheses were tested. They should incorporate some correction (e.g. Benjamini-Hochberg) for false discovery rate in the setting of 16+ hypotheses being tested. My suspicion is that the corrected p-value for chemotherapy only will be above 0.05.
Response 3: Thank you for your observations. Based on your comments, we improved the presentation of the data in Table 1. We placed all the variables in the first column, the study population in the second column, and then the columns for patients with LAEs and without LAEs. The statistical tests used aimed to determine if there was a difference between the groups of patients who had LAEs versus patients who did not have LAEs. As you mentioned, chemotherapy alone was the only variable with a significant difference. Finally, we also modified the footer of this table, where the tests used for each variable are mentioned (page 5).
Reviewer 2 Report
Comments and Suggestions for Authors
This an interesting study that needs some improvements.
Authors should use the strobe checklist for cross-sectional study to report the study performed with quality. Add the checklist in the appendix.
Please, add the study design in the title and abstract.
Where is the sample composition flowchart?
Where is the sample power calculation? The sample is very few. The author should explain better how these results could represent this type of patient.
How many patients were excluded? What is the sampling universe?
Explain in more detail how LAE was collected and identified. Was the doctor who put it in the medical record called to confirm what he had written in the medical record?
In order to identify outcome-associated LAE, the odds ratio (OR) must be used as a measure of association in bivariate analysis and the OR of each outcome should be adjusted for sex, age, and treatment at the beginning of the follow-up using logistic regression.
There is a similar study that authors could use to discuss their results. Livinalli at al. 2019 doi: 10.1097/MD.0000000000014921
Round 2
Reviewer 2 Report
Comments and Suggestions for Authors
Authors should use the strobe checklist for cross-sectional study to report the study performed with quality. Add the checklist in the appendix.
Add the study design in the title and abstract.
Where is the sample composition flowchart?
Where is the sample calculation?
How many patients were excluded? What is the sampling universe?
Explain in more detail how AEs were collected and identified. Did you apply Naranjo for causal nexo? Was the doctor who put it in the medical record called to confirm what he had written in the medical record?
In order to identify outcome-associated LAE, the odds ratio (OR) must be used as a measure of association in bivariate analysis and the OR of each outcome should be adjusted for sex, age, and treatment at the beginning of the follow-up using logistic regression.
There is a similar study that authors could use to discuss their results. Livinalli at al. 2019 doi: 10.1097/MD.0000000000014921